# pH Responsive *Abelmoschus esculentus* Mucilage and Administration of Methotrexate: In-Vitro Antitumor and In-Vivo Toxicity Evaluation

**DOI:** 10.3390/ijms23052725

**Published:** 2022-03-01

**Authors:** Sobia Noreen, Sara Hasan, Shazia Akram Ghumman, Syed Nasir Abbas Bukhari, Bushra Ijaz, Huma Hameed, Huma Iqbal, Afeefa Aslam, Mervat Abdelaziz Mohamed Elsherif, Shazia Noureen, Hasan Ejaz

**Affiliations:** 1Institute of Chemistry, University of Sargodha, Sargodha 40100, Pakistan; ssarahhsn@gmail.com (S.H.); humaiqbal203@gmail.com (H.I.); shazianoureen11@gmail.com (S.N.); 2Department of Chemistry, Sargodha Campus, The University of Lahore, Sargodha 40100, Pakistan; 3College of Pharmacy, University of Sargodha, Sargodha 40100, Pakistan; shazia.akram@uos.edu.pk (S.A.G.); afeefa_aslam@yahoo.com (A.A.); 4Department of Pharmaceutical Chemistry, College of Pharmacy, Jouf University, Sakaka 72388, Saudi Arabia; 5Centre of Excellence in Molecular Biology, University of the Punjab, Lahore 54000, Pakistan; bijaz@cemb.edu.pk; 6IRSET, EHSEP, INSERM, University of Rennes 1, 35000 Rennes, France; huma4748@gmail.com; 7Chemistry Department, College of Science, Jouf University, Sakaka 42421, Saudi Arabia; maelsherif@ju.edu.sa; 8Department of Clinical Laboratory Sciences, College of Applied Medical Sciences, Jouf University, Sakaka 72388, Saudi Arabia; hetariq@ju.edu.sa

**Keywords:** biopolymer, anticancer drug, sustained delivery, pH-responsiveness, in-vivo toxicity, antitumor activity

## Abstract

The rapid progression in biomaterial nanotechnology apprehends the potential of non-toxic and potent polysaccharide delivery modules to overcome oral chemotherapeutic challenges. The present study is aimed to design, fabricate and characterize polysaccharide nanoparticles for methotrexate (MTX) delivery. The nanoparticles (NPs) were prepared by *Abelmoschus esculentus* mucilage (AEM) and chitosan (CS) by the modified coacervation method, followed by ultra-sonification. The NPs showed much better pharmaceutical properties with a spherical shape and smooth surface of 213.4–254.2 nm with PDI ranging between 0.279–0.485 size with entrapment efficiency varying from 42.08 ± 1.2 to 72.23 ± 2.0. The results revealed NPs to possess positive zeta potential and a low polydispersity index (PDI). The in-vitro drug release showed a sustained release of the drug up to 32 h with pH-dependence. Blank AEM -CS NPs showed no in-vivo toxicity for a time duration of 14 days, accompanied by high cytotoxic effects of optimized MTX loaded NPs against MCF-7 and MD-MBA231 cells by MTT assay. In conclusion, the findings advocated the therapeutic potential of AEM/CS NPs as an efficacious tool, offering a new perspective for pH-responsive routing of anticancer drugs with tumor cells as a target.

## 1. Introduction

The potential to attain efficacious therapeutic dosage with minimal perils has taken precedence for researchers in the recent decade. Nonetheless, the prevailing itineraries to cancer treatment, such as chemotherapy still require many refined routes to attain the best results in response to traditional drugs. The nano-scaling of chemotherapeutics whilst encapsulating them within carriers with stimuli-responsive aptitude seems to be a probable route to tackle shortcomings. The engineering of such carriers to respond to certain stimuli promises targeted delivery of anticancer drugs to tumor cells by establishing endosomal uptake and adornment with targeting agents. Such stimuli-responsive nanocarriers have the potential to trigger the release of cargo (drug) enclosed within once they reach the tumor site, in response to particular stimuli for instance pH, hypoxia, and overexpression of certain enzymes [1,2].

In this smart nanomaterials research decade, the tide has shifted largely towards biodegradable polymers, polysaccharides and lipids. Polysaccharide nanoparticles (PNPs), in addition to nano-ranged size, have the potency to procure a sustained payload release, attuning their composition to preeminent outcomes [3]. Moreover, their inherent properties, such as abundance, non-cytotoxicity, biodegradability, low cost, biocompatibility have improved their use in fabricating drug delivery modules, particularly for anticancer drugs encapsulation [4,5]. Polysaccharides, such as chitosan [6], hyaluronic acid [7], dextran [8], alginate [9], *Angelica sinensis* [10], and fucoidan [11] nanocarriers have been devised as anticancer agents and demonstrated to not only have the potential to curtail the hazards but also acquire adequate therapeutic indices with minimal side effects.

*Abelmoschus esculentus* mucilage (AEM), natively known as okra gum, is an anionic polysaccharide that includes a dominant blend of galactose, rhamnose and glucuronic acid monomers. It has been widely used as a drug excipient, binder, film-forming and drug delivery agent with an acetylation degree (DA = 58) [12]. Chitosan (CS) is an abundant cationic polysaccharide obtained by alkaline chitin deacetylation. It offers broad therapeutic applications varying from the manufacturing of food to targeted cellular payload delivery. The extensive use of CS in NPs is indebted to its cationic nature which aids in plummeting the circulation period and amplifying bioavailability upon biological environment exposure [13]. Furthermore, it shows the excellent permeability of encapsulated drugs due to its ability to open tight junctions between epithelial cells [14]. The efficacy of CS-NPs is often improved by their tailoring with anionic polysaccharides that have shown to decrease their macrophagal uptake, enhancing the efficacy of the encapsulated cargo [15]. Moreover, both AEM and CS polymer-based nanocarriers are known to give effective protection against gastrointestinal degradation. As a result, they increase bio-sorption owing to their bio-adhesive nature, encouraging their use in site-specific delivery devices [16,17]. For instance, AEMCS NPs have been found to possess an excellent mucoadhesive character with targeted delivery of esculin. Likewise, thiolated AEM-NPs have proven to be a compelling drug carrier with target precision to the brain conceding its potential in delivery systems [18].

Methotrexate (MTX), a folic acid antagonist has been in clinical instrumental use for cancer and autoimmune diseases. It inhibits folic acid metabolism by blocking the activity of dihydrofolate reductase by interfering in the de novo synthesis of DNA, RNA and proteins [19,20]. However, its therapeutic efficacy is often seriously compromised by its insignificant targeting tendency that often leads to side effects, such as alopecia, nausea, body aches, hepatotoxicity and myelosuppression [21]. The poor targeting capability and low bioavailability of MTX can be boosted by its encapsulation within nanocarriers [22]. Although, in the past decade, several natural polysaccharides based nanocarriers for MTX targeted delivery have been developed. For instance, phytic acid and CS-based NPs by ionic gelation for MTX controlled release, were formulated by Ciro et al., [23]. Wang et al., (2021) prepared lactobionic modified thymine and CS comprised nanocarriers for MTX [24]. A biodegradable nanocarrier based on CS and silica was synthesized by Shakeran et al., for the treatment of breast cancer [25]. Bhattacharya formulated polymeric lipid hybrid NPs for glioma treatment by MTX [26]. However, the preparation steps involved in such NPs are complex and most of these require the addition of a cross-linker. Additionally, such delivery modules require organic solvents in their synthesis pathway posing toxicity and protein denaturation. In an attempt to selectively deliver MTX payload to tumor physiological environment, with minimal toxicity level, to normal cells we prepared AEM-CS NPs by a slightly modified coacervation method. The method utilized was simple as it allowed self-crosslinking of polymeric matrix followed by electrostatic interactions of MTX. The characterizations of the resulting nanoparticles regarded size, shape and encapsulation efficiency. MTX release profiles at gastrointestinal pH (1.2 and 7.4) and tumor pH (5.5) were studied to ensure the targeting capability of NPs as pH-responsive nanocarriers. We also exposed MCF-7 and MD-MBA231 cell lines and non-cancerous Vero cells to MTX/AEM-CS NPs for cell viability evaluation which showed that MTX/AEM-CS NPs could be a potential alternative to commercial anticancer drugs, alongside the provision of side-effect free chemotherapeutics.

## 2. Results and Discussion

Low bioavailability and interaction with normal cells are the major concerns associated with cancer therapeutics. A solution to these inadequacies seems to be the utilization of encapsulation of cancer drugs with natural polymers with efficient methods. AEM and CS were chosen as carriers for MTX delivery because of their safe, biodegradable and biocompatible nature.

Inter-chain bonding between AEM (anionic polymer) and CS (cationic polymer) led to the formation of electrostatic interactions, demonstrated in Figure 1. Moreover, weak electrolyte formation between CS and AEM led to a lowering of the free system energy which augmented the overall complex stability as discussed in another study [27].

The present study was carried out with the aim that AEM-CS based formulations would enable the chemotherapeutics to be used to their full therapeutic potential in controlled ways whilst alleviating the toxicities associated with them. Moreover, the encapsulation of MTX with AEM mucilage would enable these nanoparticles to be delivered orally as it would provide protection from the gastric environment. For it, several formulations (F1, F2, F3) were prepared with varying concentrations of AEM and CS in ratios of 2:1, 1:1, 1:2, respectively. Methotrexate was used as a model drug at a concentration of 1 µg/mL.

### 2.1. FTIR Spectroscopy

The two charged polymers interacted to form a polyelectrolyte complex which was characterized by FTIR. The absorption peaks of MTX shown in Figure 2d at 3766.98 cm^−1^ and 3564.45 cm^−1^ indicated imines and O-H stretching vibration, 3323.35 cm^−1^ for the aromatic group, 2937 cm^−1^ for alkyl groups, 1870.95 cm^−1^ for carboxyl group stretching vibration. Medium peaks for amide C=O stretching at 1680 cm^−1^, and C=C at 1492.9 cm^−1^ in the aromatic ring were observed in the drug methotrexate. The amine group shows its presence at 952.84–1224.8 cm^−1^. C-C stretching is present at 817.82 cm^−1^ [28,29].

The spectrum of CS (Figure 2b) shows character peaks at 3325.28 cm^−1^ and 2862.36 cm^−1^ which represent the –OH and -CH2 groups, respectively. The peaks at 1527.62 and 1456.26 cm^−1^ show the NH- bending vibration and the –OH group of primary alcohol. The peak at 1031.92 represents C-O stretching [30].

The AEM spectrum (Figure 2c) shows peaks at 3378.90 and 2937.69 cm^−1^ of stretching vibration of NH2 and OH. The presence of –OH represents the hydrophilic character of polymer. The band at 2478.53 cm^−1^ is due to the CH stretch of –CH_3_. The peak at 1741.72 cm^−1^ is a C=O stretch that can be found in galacturonic acid; 1222.8–869.9 cm^−1^ was the fingerprint region of carbohydrates [31].

The IR spectra of MTX loaded AEM-CS NPs in Figure 2a show a shift in the COO-peak shift to 1867.45 cm^−1^, which is indicative of electrostatic bond formation between two AEM and CS. The bands at 3417.01 and 2997.46 cm^−1^ are representative of –NH2 and hydrogen bonded –OH present in CS too. Additionally, the characteristic peaks for cyclic alcohols can be seen in the regions of 1200–1000 cm^−1^, and are visible in the spectra of polymers too. The spectra indicated that no interaction occurs between the functionalities of the physical mixture of MTX, AEM and CS.

### 2.2. Entrapment Efficiency

The % entrapment efficiency (% EE) of MTX-loaded AEM-CS NPs formulations was found to be in the range from 42.08 ± 1.2 to 72.23 ± 2.0. The % EE was observed to be decreasing with an increase in either of the polymer’s concentrations. This is mainly due to the formation of a loose polymeric network when either of polymer concentrations is increased leading to drug leaching from the matrix, the same results have been observed by studies carried out by Kajjari et al., whilst working on chitosan-guar gum NPs for ciprofloxacin release [32]. The highest % EE was observed in formulation F2 (1:1) due to adequate interactions between AEM and CS, causing the formation of a compact polymeric matrix which in turn caused a surge in the drug entrapment efficiency of NPs (Table 1).

### 2.3. % Drug Content and Percent Yield

The % drug content of MTX-loaded AEM-CS NPs was found to be in the range of 81.2 ± 1.2 to 94.5 ± 1.6 (Table 2). It was observed that the formulation with a 1:1 AEM-CS ratio, i.e., F2 showed an increase in % drug content. However, in F1 and F3 with AEM-CS ratios of 2:1 and 1:2, respectively, the drug content was decreased. This is due to an increase in size which causes the surface area of NPs to be decreased, leading to drug content lowering. The % yield was found to be improved, ranging from 50 ± 1.2 to 84.3 ± 0.8 with an increase in polymer ratios. The increase in yield is due to an increase in the NPs weight owing to the increase in concentrations of AEM and CS, though the increase was more pronounced in F3, with an AEM-CS ratio of 2:1, which is in accordance with previously published research [33,34].

### 2.4. Particle Size Distribution, Zeta Potential and Morphology of AEM-CS Nanoparticles

The size distribution and ζ-potential values of AEM-CS nanoparticles are described in Table 3. Dynamic light scattering technique is used to determine the size distribution profile of nanoparticles [35]. Results revealed that the average hydrodynamic diameter of nanoparticles formulations was in the range of 211–269 nm with an acceptable PDI range of 0.279–0.485. Particle size was smallest when the AEM-CS ratio was 1:1 (Figure 3A). The polydispersity index (PDI) is a dimensionless parameter that tells us the heterogeneity in the dispersion of detected particle size. PDI values less than 0.1 are considered as “Monodisperse” and PDI values greater than 0.7 are considered as “Polydisperse” [36]. PDI results were calculated by data obtained from DLS analysis. Particle size was smallest when the AEM-CS ratio was 1:1 (Figure 3A,C). Changes in the mass ratio of CS and AEM resulted in a shift in particle size. An increase in either CS or AEM leads to bigger particle size. These findings indicated that the AEM-CS ratio affects the particle size [35].

Pure chitosan is a cationic polysaccharide due to the presence of free amino groups, whereas pure AEM is anionic because of free carboxylic groups [37,38]. These groups are responsible for the net positive or negative charge of polymers and influence ζ-potential values. Correspondingly, for nanoparticles prepared with equal mass ratios of CS and AEM, the ζ-potential of nanoparticles was +11.4 mV. An increase in ζ-potential value was observed when CS mass ratio was increased. On the other hand, ζ-potential was reduced upon increasing the ratio of AEM in composition. ζ-potential values of nanoparticles prepared with different AEM-CS ratios are shown in Figure 3B. In Figure 3C, the SEM images along with their respective histogram representing the size distribution of nanoparticles formulations (F1-F3) of three different ratios were presented, which revealed that all nanoparticles had solid, smooth, and spherical shapes. By applying the Gaussian fit of counts on distribution frequency, the mean size distribution was in accordance with DLS analysis. DLS analysis only talks about size distribution but the exact shape of nanoparticles accessed with SEM analysis and more even size has been clearly seen at a ratio of 1:1 in Figure 3C.

### 2.5. In-Vitro Drug Release Studies

A tumor targetted drug delivery system requires the exhibition of little or no drug release before its site of action (cancerous cells). If not, then the drug will be absorbed in the gastrointestinal tract (GIT) causing a decrease in the efficacy of the drug. Additionally, the sustained and controlled delivery of hydrophobic drugs is preferred as it reduces the need for frequent drug administration, besides decreasing several side effects [39,40].

The MTX cumulative release from AEM-CS nanoparticles in simulated normal and cancerous cells conditions was studied, to determine the pH responsiveness of formulated NPs towards the cancerous microenvironment (Figure 4). The release profiles in simulated gastric fluid (SGF; pH 1.2) for the first 2 h and later in simulated intestinal fluid (SIF; pH 7.4) for 30 h were carried out. This was performed to mimic the pH and transit time of the gastrointestinal tract (GIT). At pH 1.2, all formulations (F1, F2, F3) showed an insignificant MTX release in between 12.13–14.01% range while at pH 7.4 CDR of 41.62–53.4% was observed, respectively. The quantity of drug discharge was increased with an increase in polymer ratio because of the enhancement in the diameter of the polymeric membrane and poor interaction between polymers which surged the diffusion rate of the drug in GIT mimicking media [32,41].

At pH 5.5, the release rates of formulation (F1, F2, F3) showed an escalated rate of 53.17%, 60.91% and 42.76%, respectively. The drug was released more quickly at pH 5.5 due to the protonation of carboxyl and amino groups present on the AEM-CS surface which increased the swelling potential of both polymers (AEM and CS) in a slightly acidic medium. This is due to the repulsion between similar charges that allowed faster drug release in pH 5.5 [42]. In contrast, the release is slow in pH 1.2 and 7.4 as the swelling index is low. The results were consistent with other previous studies carried out on the pH-responsive nanocarriers of MTX though unlike our NPs most of these require a hectic synthesis procedure [42,43,44].

The F2 nanoparticles, which contain 0.01 *w*/*v*% AEM and 0.01 *w*/*v*%, exhibited better sustained drug release compared to other formulations, showing a more sensitive and efficient drug release at pH 5.5 and slow sustained release at pH 1.2 and 7.4, indicating nanoparticles sensitivity to tumor pH [45]. This efficient MTX release at acidic tumor pH from spherical AEM-CS NPs prevents the loss of drug in the vicinity of normal cells with improved endosomal MTX uptake [46]. The results advocate the value of these tumor pH-responsive nanoparticles as effective means to release chemotherapeutics in the tumor environment reducing the damage to normal tissues.

### 2.6. Drug Release Kinetics

To investigate the MTX release kinetics from AEM-CS nanoparticles, results from five mathematical models were analyzed. The selection of kinetic model was based on R^2^ obtained close to unity. Table 4 shows MTX release kinetic constant (k) and correlation coefficient (R^2^). At all pH values, i.e., 1.2, 5.5 and pH 7.4, the nanoparticles showed a good fit to the Higuchi model. The Higuchi model has also been used in multiple published studies to investigate anticancer drug release from biopolymer-based carriers [47,48]. This model comprises three suppositions. Initially, the drug content is higher than drug solubility potential, accompanied by the fact that particle size is less than matrix thickness with no drug particle on the surface. Lastly, the diffusion of the drug is constant and mainly occurs via pores hydrophilic polymeric matrix solubilizes easily on contact with solvents [49].

The data was further fitted to Korsmeyer Peppas and Peppas–Sahlin model to understand the release mechanism. The said models indicated that the drug release was driven by diffusion and polymer chain relaxation in the case of pH 5.5. In this scenario, the diffusion dominates in accord with K_d_ and K_r_ values as the K_d_/K_r_ ratio is >1. The “n” values range between 0.43 and 0. indicates an anomalous diffusion drug release too (Table 4a) [50].

The kinetic release results at pH 1.2 and 7.4 showed that the diffusion phenomenon was mainly followed during the drug release. The low n values, i.e., <0.45 calculated by the Korsmeyer Peppas model and K_d_/K_r_ in Peppas–Sahlin affirmed the mediation of drug release by diffusion which is in agreement with published literature (Table 4b) [50,51,52].

### 2.7. In-Vivo Acute Toxicity of AEM-CS Based Blank Nanoparticles

Acute toxicity is used to assess the toxicity of drug delivery devices and their lethal dosage (LD50) values. Furthermore, the use of relevant species of animals for in-vivo toxicity analysis has a principal role in evaluating the long-term toxicity data. These studies can ultimately be translated into accessing the human tissue’s reactivity profile [53,54]. In the present study, neither any toxic effects (skin color alteration, tremors, and diarrhea) nor mortality were observed. Moreover, the behavioral and sleep patterns remained the same, and no mortality was observed in mice for a time duration of 14 days. No significant body weight change was observed on either of the groups for 14 days and since no animal died, the lethal dosage could not be measured. The food and water consumption were the same in both groups, though a slight weight decrease was observed in the test group. However, this weight change was not different when compared to control group animals (Figure 5a).

### 2.8. Blood and Hematological Analysis

Furthermore, hematological and biochemical profiles were assessed (Figure 5), as these parameters facilitated estimating the influence of the drug-delivery system on blood composition. The results depicted no substantial alteration in hematological and biochemical profiles of the tested group in comparison to the control one.

Furthermore, the liver and kidneys are the main sites for biological parameters fluctuations and damage to such organs, is most likely to increase the hepatic enzymes (ALT, AST and ALP) and renal creatinine. The analysis of these hepatic and renal parameters demonstrated the AEM and CS constituted NPs to be non-toxic and safe. Besides that, no inflammation and degeneration in the liver and kidneys of mice were observed, prompting the use of the AEM-CS NPs based drug carriers as chemotherapeutics delivery modules [55,56].

### 2.9. Cytotoxicity Analysis

The in vitro cytotoxic activity of MTX loaded NPs (F2) on normal cells (Vero cell lines) and cancer cell lines (MCF-7 and MDA-MB231) was investigated by MTT assay. The cell viability was significantly high for Vero cells on treatment with blank AEM-CS NPs, showing their non-toxicity towards these cells and significant biocompatibility.

Formulation F2, with an AEM and CS ratio of 1:1, was selected as a potential candidate for this study as it has the smallest size and the highest drug load amongst all three formulations. Additionally, the blood capillaries are of diameter 5–6 um, so the administration of smaller sized NPs can help in the prevention of embolism and capillaries blockage. Vero cells were less sensitive towards the anti-proliferating and toxic activity of MTX loaded AEM-CS NPs. Therefore, even after an incubation time duration of 48 h MTX and MTX loaded AEM-CS NPs did not substantially cause cell death, i.e., cell viabilities with minimalistic values of 75% and 78.02%, respectively.

The MTX and MTX loaded AEM-CS NPs (Figure 6B) significantly inhibited cell proliferation in both MCF-7 and MDA-MB231 cancer cells after 48 h, whereas there were no significant cytotoxic effects on cell viability in the case of the Vero cell line [56]. We observed a high time and concentration dependent activity in MTX loaded AEM-CS NPs in comparison to free MTX in concentrations of 3.12–200 µg/mL, It was observed that in the initial 24 h the MTX loaded AEM-CS NPs displayed a slightly higher % cell viability ratio than free MTX (Figure 6A). This can be evidenced by the fact that free drug is rapidly taken up by cancerous cells via passive diffusion due to the cell’s immediate exposure to whole drug concentration whereas only a portion of the drug is exposed to cell lines in NPs [57]. The drug is expected to gradually increase till it attains a steady state in the NPs. Therefore, after 48 h, a remarked high cellular cytotoxicity for both cell lines was observed by MTX loaded NPs [58,59]. The anticancer efficacy, however, was significant when compared with several previously reported nanocarriers for MTX. However, further in-vivo studies are required for the complete characterization of anticancer efficacy of MTX loaded AEM-CS NPs.

### 2.10. Stability Studies

The particle size and entrapment efficiency of MTX-loaded NPs were determined initially and later weekly till 21 days, shown in Figure 7. The data showed no potential difference in the size of nanoparticles and encapsulation efficiency of formulation (F2), depicting the NPs encapsulation layer stability.

## 3. Material and Methods

### 3.1. Material

Chitosan (medium molecular weight, 75–85% de-acylated) was purchased from Sigma Aldrich. Methotrexate was received as a gift from the pharmaceutical company of Peshawar. Ethanol, sodium hydroxide, glacial acetic acid >99.7%, and other chemicals and reagents used in the present study were obtained from Sigma-Aldrich (GmBh chemie, Schnelldorf, Germany) and Fisher Scientific, UK. All solutions preparation was made by using ultrapure water and all chemicals utilized were of analytical grade.

### 3.2. Mucilage Extraction

Fresh Okra pods were purchased from the local market, Sargodha, Punjab, Pakistan. The authentication of the plant was performed at the Department of Botany, University of Sargodha, Sargodha. Mucilage extraction was performed by following reported methods by Baveja and Wahi [60,61]. Fresh pods were washed and cut into small pieces. The mucilage precipitation was carried out by ethanol in water in ratio 3:1. Once separated, the mucilage was oven-dried at 40 °C and passed through sieve #80 and stored in airtight vials for future study. Mucilage suspension was prepared by adding it into deionized water followed by centrifugation for 20 min at 3000 rpm at room temperature. The supernatant was evaporated, and mucilage was obtained as the pellet was freeze-dried, and stored in a plastic air-tight container.

### 3.3. Formulation of Nanoparticles

Methotrexate nanoparticles were prepared with the help of an earlier reported coacervation method with slight modifications followed by ultra-sonification [62,63]. A solution of AEM (0.02% *w/v*) in distilled water was prepared under constant stirring with a pH adjustment of 5.2 by 0.1N HCl. Whereas the preparation of CS (0.02% *w/v*) was performed in 0.1% acetic acid under magnetic stirring. The pH of the CS solution was adjusted to 5.5 by using 5.5 1N NaOH.

AEM solution was added to the CS solution with a stirring duration of 20 min at 40 °C. MTX solution (1 µg/)mL was prepared by using dimethyl sulfoxide (DMSO). For this, 10 mg of MTX was dissolved in 10 mL of DMSO and serially diluted further to obtain a solution of 1 µg/ mL concentration. Three formulations were prepared in a similar way altering the polymers (CS and AEM) ratios as 2:1, 1:1, 1:2 while keeping the drug (MTX) concentration constant (1 µg/mL). The samples obtained in the result were centrifuged at 12,000 rpm at 40 °C for 45 min. The NPs obtained in form of coacervate were lyophilized and stored.

### 3.4. Fourier Transform Infrared (FTIR) Spectroscopy

Spectral investigations of AEM-CS nanoparticles, native polymer (AEM and CS) and drug (MTX) were recorded in the region 4000–400 cm^−1^ by IR Prestige-21 spectrophotometer (Shimadzu, Kyoto, Japan). This analysis was made for the identification of any interaction between polymers and drugs [64,65].

### 3.5. Nanoparticles Evaluation

#### 3.5.1. Dynamic Light Scattering (DLS) Analysis

The particle size, polydispersity index (PDI) and zeta potential evaluation of MTX-AEM-CS NPs, with 1 mg/mL concentration, was performed by a Zetasizer Nano ZS (Malvern Instruments, Malvern, UK) at room temperature [66]. For this investigation, dilute solutions were employed, as concentrated ones lead to false measurements. The inaccuracy in particle size study in concentrated solutions is mainly owed to the multiple scattering [67]. Cumulants method of analysis was used in Malvern software to analyze the data. With appropriate refractive indices for the bulk suspension droplet, this software considered each particle as a sphere and considered that in bulk distribution. To avoid the tendency of particle aggregation, a 0.2 μm syringe filter was used to add the samples. Each formulation was analyzed thrice and the collected results were averaged. Data obtained from the same software was also used for the analysis polydispersity index [68]. Polydispersity index is calculated by dividing the square of standard deviation with average particle diameter by Equation (1).
PDI = (σ^2^/d)(1)

#### 3.5.2. Morphological Analysis

The analysis of morphology and surface of optimized MTX loaded AEM-CS NPs was made by FEG–SEM EM8100F (Field Emission Gun Scanning Electron Microscope equipped with EDX Analyzer) with an accelerating voltage of 20 kV. Image J software was used for the analysis of SEM micrographs. The samples were placed on double-sided tape secured on aluminum stubs and scanned at 15 KV after gold coating [68,69]. The micrographs were acquired after adding one drop of NPs suspension over copper grid accompanied with negative attaining by uranyl acetate at 60 kV accelerating voltage [58].

#### 3.5.3. Encapsulation Efficiency

For the determination of encapsulation efficiency (EE), all the formulations were suspended in water; 0.5 mL and DMSO; 0.9 mL based binary mixture. The suspensions were centrifuged at 14,000 rpm for 5 min at 4 °C in a centrifuge machine to assess the MTX amount. The supernatant obtained was filtered by Whatman filter paper and the amount of un-incorporated MTX was quantified from the absorbance in UV-spectrophotometer from Shimadzu (Kyoto, Japan) in a scanning range of 200–400 nm [65,70].

The EE of polymers in formulations was calculated by means of the calibration curve of standard MTX solutions (10, 20, 40, 60, 80, 100 µg/mL) at 303 nm by Equation (2).
(2)$$\mathrm{Encapsulation}\;\mathrm{efficiency}=\frac{\mathrm{Amount}\;\mathrm{of}\;\mathrm{total}\;\mathrm{drug}-\mathrm{Amount}\;\mathrm{of}\;\mathrm{free}\;\mathrm{drug}}{\mathrm{Amount}\;\mathrm{of}\;\mathrm{total}\;\mathrm{drug}\;\mathrm{added}}\times 100$$

#### 3.5.4. Percentage Yield and Drug Content

For percentage drug content, 5 mg of MTX loaded AEM-CS NPs were suspended in water (0.5 mL) followed by the addition of DMSO (9.5 mL). The nanoparticle suspensions were centrifuged at 12,000 rpm for 45 min under the cooling environment same as in the refrigerator (5 °C). The supernatant was collected, and the NP pellets were dried up. The concentration of MTX was determined by UV-Spectrophotometer (Shimadzu, Kyoto, Japan). An equal amount of AEM-CS blank NPs was also used as blank (Equation (3)). For calculation of % yield equations 4, the weights of nanoparticles, pure drug and polymers were also determined [65,71].
(3)$$\%\;Drug\;content=\frac{\mathrm{Weight}\;\mathrm{of}\;\mathrm{drug}\;\mathrm{in}\;\mathrm{nanoparticles}}{\mathrm{Weight}\;\mathrm{of}\;\mathrm{nanoparticles}\;\mathrm{recovered}}\times 100$$
(4)$$\%\;Yield=\frac{\mathrm{Weight}\;\mathrm{of}\;\mathrm{nanoparticles}\;\mathrm{recovered}}{\mathrm{Weight}\;\mathrm{of}\;\mathrm{total}\;\mathrm{solid}\;\left(\mathrm{AE}+\mathrm{CS}+\mathrm{MTX}\right)}\times 100$$

#### 3.5.5. In-Vitro Drug Release

The in-vitro MTX release was determined by the dialysis sac method via dissolution rate apparatus, USP type-II (PTD7, Pharma-Test, Hainburg, Germany). A 3 mL sample of NPs was placed in a dialysis sac (cut off 10,000 kDa) and secured to paddle with a release medium of 200 mL of phosphate buffer at pH 1.2 for an initial 2 h which was later replaced by alkaline phosphate buffer solution (PBS) of pH 7.4 with constant stirring at 25 rpm at 37 ± 0.5 °C. The procedure was repeated for all formulations at pH 5.5 so as to study the release rates in the tumor microenvironment. A 3 mL sample was withdrawn from cells at 30, 45, 1, 2, 3, 4, 5, 6, 8, 10, 12, 24, 28 and 32 h, respectively whilst replacing it with an equivalent volume of PBS. The absorbance of withdrawn samples was noted with UV-spectrophotometer at 303 nm. The concentration at a given time (t) was obtained by the calibration curve of MTX. The standard calibration curve of methotrexate in PBS (1, 2, 3, 4, 6, 8, 10, 15, 20, 25, 30 µg/)mL was obtained and release was determined by Equation (5) [45,72].
(5)$$\mathrm{Drug}\;\mathrm{release}\;\%=\frac{\mathrm{Amount}\;\mathrm{of}\;\mathrm{MXT}\;\mathrm{released}}{\mathrm{Amount}\;\mathrm{of}\;\mathrm{loaded}\;\mathrm{MTX}\;}\times 100$$

#### 3.5.6. Kinetic Studies of Drug Release

The dissolution profiling of MTX loaded AEM-CS NPs was carried out by kinetic models to ensure that the drug dissolution from nanoparticles was occurring in an appropriate manner [70], as follows:

Zero Order model: The initial use of this model was mainly to describe the MTX-loaded AEM-CS NPs at a concentration-independent release rate, represented by Equation (6).
(6)$$Q_{t}=Q_{0}+K_{0}\;\cdot \;t$$where $Q_{t}$= amount of drug dissolved at the time ‘*t*’; *Q_0_* = Initial drug amount in solution.

First Order model: The model explored the release of hydrophilic drugs from a porous matrix in comparison to drug content inside the drug carrier, in accordance with Equation (7).
(7)$$LogQ_{t}=LogQ_{0}-\frac{K_{1}}{2.303}\;t$$where $Q_{t}\;$= amount of drug dissolved at time t; *Q_0_* = Initial drug amount in solution; *K*_1_= first-order rate constant.

Higuchi model: This model was used to describe the cumulative percent release of hydrophilic drugs from hydrophobic polymeric matrixes in relation to the square root of time, represented by Equation (8).
(8)$$Q_{t}=K_{H}\;t^{\frac{1}{2}}$$

*K_H_* = Higuchi model constant; *Q_t_* = amount of drug dissolved at time ‘*t*’

Korsmeyer–Peppas model: The characterization of the mechanism involved for cumulative drug release incorporated within polymeric matrixes was made by this model, Equation (9).
(9)$$\frac{M_{t}}{M_{\infty }}=K_{r}t^{n}$$

$\frac{M_{t}}{M_{\infty }}$ = drug fraction release at time “*t*”; $K_{r}$ = release constant; *n* = release mechanism dependent exponent

Peppas–Sahlin model: The extent of drug diffusion accompanied by polymeric matrix relaxation was described by the Peppas–Sahlin kinetic model, expressed by Equation (10):(10)$$\frac{M_{t}}{M_{\infty }}=K_{d}t^{0.5}+K_{r}t^{1}$$

$\frac{M_{t}}{M_{\infty }}$ = Fraction of drug with release time ‘t’; *K_d_* = rate constant of diffusion; *K_r_* = rate constant of relaxation

#### 3.5.7. In-Vivo Acute Toxicity of AEM-CS Based Blank Nanoparticles

The acute toxicity of both polymers, i.e., AEM and CS was tested by repeated dose (14 days) technique. All the experimental work was performed in accordance with OECD guidelines and verification was made by the ethics committee of the University of Sargodha (UOS), Sargodha, Pakistan under Ref. No 70B18 IAEC/UOS (PREC). For this, Swiss albino mice weighing 29 ± 1.2 g were purchased from Animal Centre, UOS, of either sex and placed in a clean house facility.

Animals were randomly distributed into two groups (*n* = 5) with Group I as control and Group II as a test group. The control group was treated with water and laboratory chow diet whilst the test group was intravenously administrated with an AEM-CS NPs dose of 10 mg/kg via tail vein for 14 days. A daily observation was made to check the signs of ill health for a time duration of 14 days.

##### Physical Observations

A daily observation was made about the health of mice, side effects in response to treatment and changes in the vitals, i.e., skin, eyes, mucosal membranes, behavior, sleep pattern and deaths for a time duration of 14 days. The body weights were also assessed before the treatment and on the 1st, 7th and 14th day after dose administration. All the measurements were compared with the control group.

##### Biochemical and Hematological Profiling

Bodyweight was measured before treatment and then observations were made on 1st, 7th and 14th along with hematological and biochemical profiling after 14 days. For biochemical and hematological profiling, the mice were anesthetized and blood samples were collected by cardiac puncture. The collected samples were analyzed for Hemoglobin; Hb, WBCs; White blood cells and RBCs; Red Blood cells, the liver enzymes (ALT; Alanine Aminotransferase, AST; Aspartate Aminotransferase, ALP; Alkaline Phosphatase) and creatinine were also analyzed according to the International Federation of Clinical Chemistry and Laboratory Medicine (IFCC) primary reference procedures using an Olympus AU2700 Chemistry analyser^®^ (Olympus Optical, Tokyo, Japan) [55].

#### 3.5.8. Cell Viability Studies

The cell viability analysis is a key study to comprehend a new drug delivery module and assessment of their biomedical applications. The in-vitro cytotoxicity of optimized nanoparticles formulation (F2) in the MCF-7 and MDA-MB231 cancerous cell lines and Vero cells taken as normal was evaluated by MTT assay. The cell lines were cultured in DMEM (Dulbecco’s Modi-field Eagle medium that contained four times higher concentrations of amino acids and vitamins) and 10% FBS. The cells were seeded at 37 °C in a 5% CO_2_ atmosphere in 96 well plates (1 × 10^4^ cells in each well) and incubated for 48 h. The test samples were solubilized in dimethyl sulfoxide (DMSO) in 500 µg/ mL concentration and diluted to 200 µg/mL by water and frozen for later use. The frozen concentration (200 µg/mL) fter being thawed, was diluted in sequential concentrations of nanoparticles (3.12, 6.25, 12.5, 5, 100 µg/mL) and incubated for 48 h with culture plates. The cell lines were also treated with the same concentration of MTX. One hundred microliters of MTT with a concentration of 5 mg/mL was added to each well, followed by additional 4 h incubation, with subsequent removal of the culture media with DMSO in each well. The absorbance was read at 570 and 650 nm via a microplate reader (Thermo Fisher Scientific, Rockfold, IL, USA) [51]. The experiment was conducted in triplicate and results are presented in mean ± S.D.
Cell viability = Test cells(abs)/Control cells(abs) × 100(11)

#### 3.5.9. Stability Studies

The stability of MTX loaded AEM-CS nanoparticles was determined at 25 °C ± 2 °C by measuring the size and encapsulation efficiency on 1st, 7th, and 14th days whilst keeping an eye on the growth of crystals if any [27].

### 3.6. Statistical Analysis

The results were analyzed by one-way ANOVA *p* < 0.05 statistically with results taken in triplicate and presented as mean ± S.D, alongside a Tukey posthoc test to evaluate the significant variations between independent variables, if noted. GraphPad Prism 8.0.2 was used for the analysis of data.

## 4. Conclusions

The present study was aimed at the successful nanoencapsulation of MTX into AEM-CS nanoparticles by using the coacervation technique. The encapsulation of MTX in the AEM-CS complex was affirmed by FTIR with nano-size validation by SEM. The study concludes that a developed drug delivery system helps to acquire the efficacious release of the cancer drug model, i.e., MTX. The fabricated nanoparticles showed appreciable encapsulation efficiency and potential to control MTX release till 32 h, especially at tumor pH. Moreover, prepared NPs exhibited substantial cytotoxic results in MCF-7 and MD-MBA231 cancer cell lines. Therefore, the present research provides an idea that the utilization of natural polymers for chemotherapeutics encapsulation can provide a gateway to overcome the side effects associated with cancer drugs with a heightened response to tumor pH, so based on our findings we can say that MTX loaded AEM-CS NPs can be a beneficial route for targeting cancerous cells and that mucilage based NPs could be a striking area for future studies in tumor therapy.

## Figures and Tables

**Figure 1 ijms-23-02725-f001:**
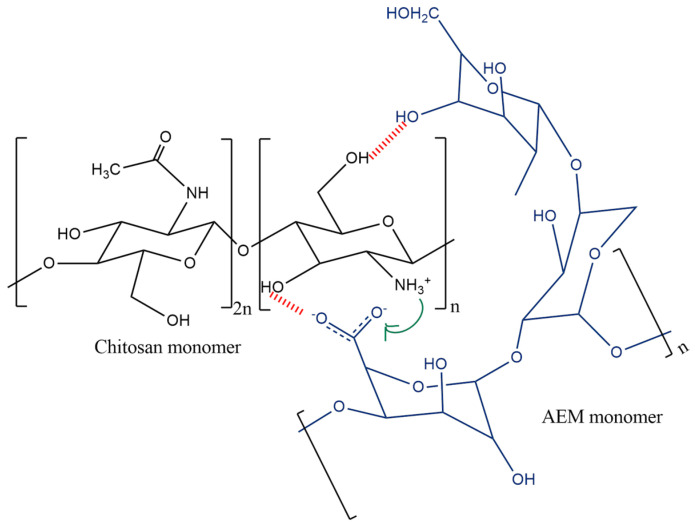
Electrostatic Interaction between AEM and CS monomers.

**Figure 2 ijms-23-02725-f002:**
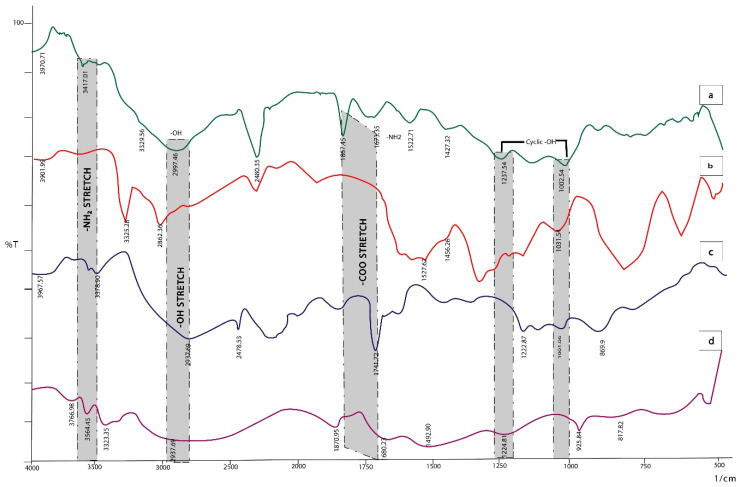
Drug polymer compatibility FTIR studies (**a**): MTX loaded AEM-CS NPs, (**b**): CS, (**c**): AEM and (**d**): MTX.

**Figure 3 ijms-23-02725-f003:**
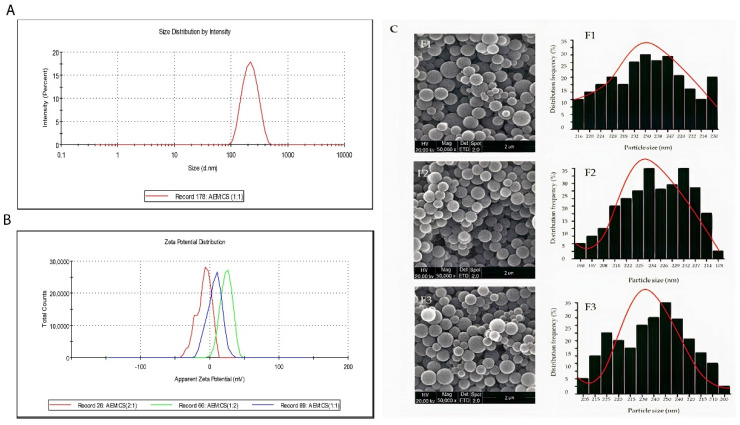
(**A**) Z-average d. nm AEM:CS at 1:1 ratio. (**B**) Zeta-potential (mV) of AEM:CS at the ratios of (2:1, 1:1 and 1:2). (**C**) SEM images along with their respective size distribution histograms of AEM:CS at the ratios of (2:1, 1:1 and 1:2).

**Figure 4 ijms-23-02725-f004:**
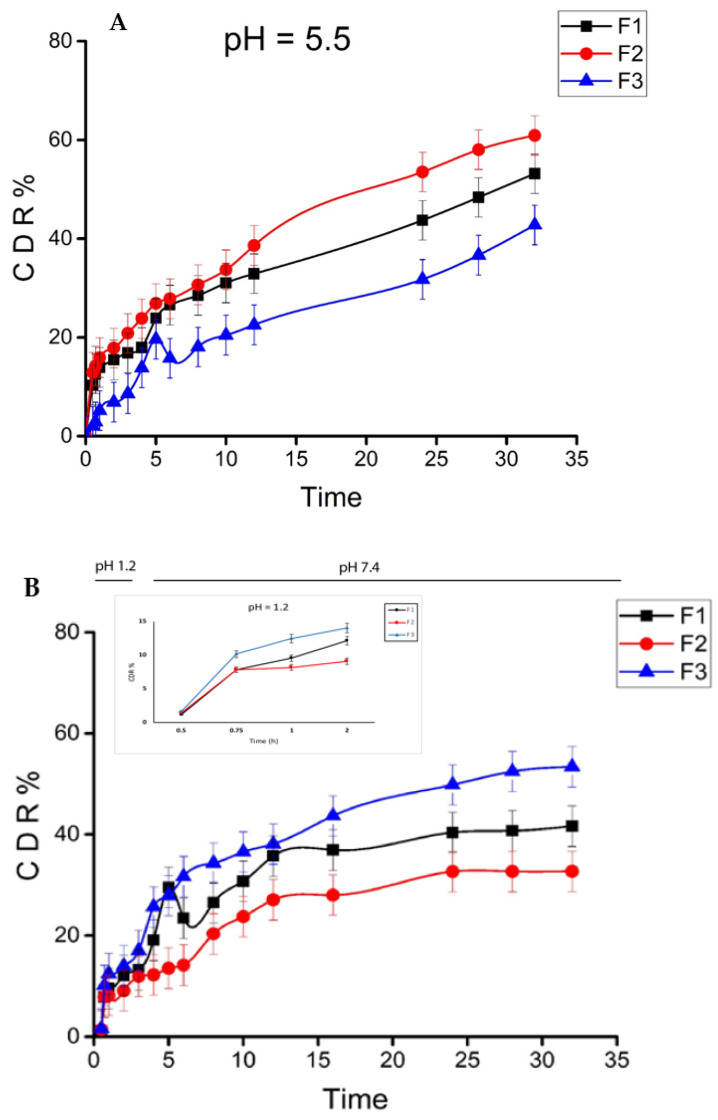
In vitro cumulative drug release percentage (CDR %) of MTX loaded AEM-CS nanoparticles. (**A**) pH 5.5, (**B**) 1.2 and 7.4 at 37 °C. (*n* = 3/ Mean ± S.D.).

**Figure 5 ijms-23-02725-f005:**
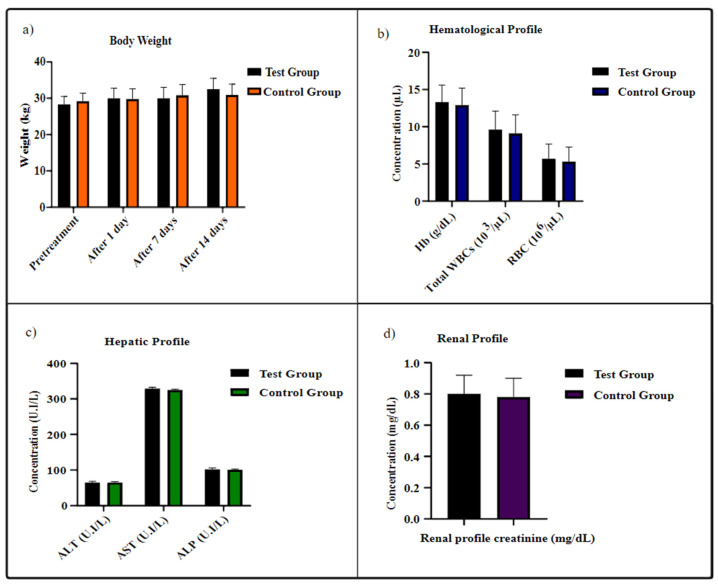
Biochemical profile of albino mice after repeated dose administration over a time duration of 14 days (**a**) body weight, (**b**) hematological profile, (**c**) hepatic profile, (**d**) renal profile Results are represented as mean ± S.D. with *n* = 7 and *p* < 0.05.

**Figure 6 ijms-23-02725-f006:**
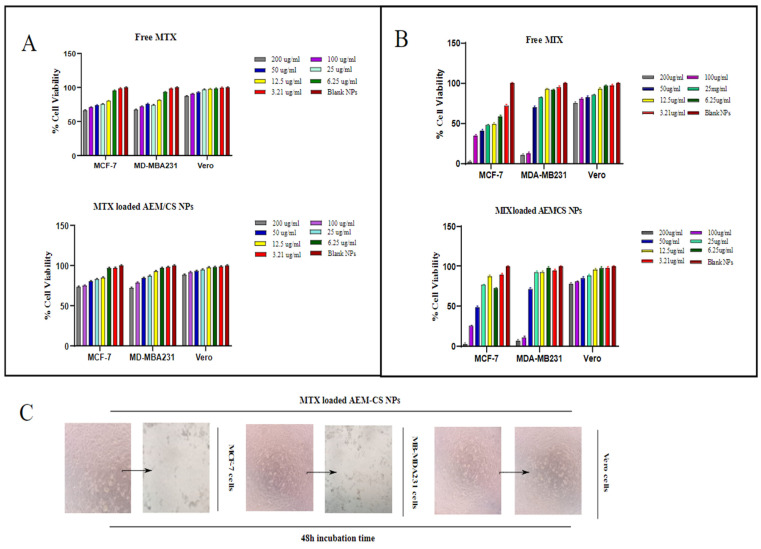
Quantitative cytotoxicity analysis of free MTX and MTX loaded AE/CS NPs (F2) on MCF-7, MDA-MB231 cancer cells and Vero (normal cells) after (**A**) 24 h, (**B**) 48 h treatment duration with DMSO and Blank AEM-CS NPs (no drug) as control, (**C**) Representative images of MCF-7, MDA-MB231 cancer cells and Vero (normal cells) upon treatment with F2 formulation (Scale bar = 1 mm). Data are presented as Mean ± S.D; *p* < 0.001.

**Figure 7 ijms-23-02725-f007:**
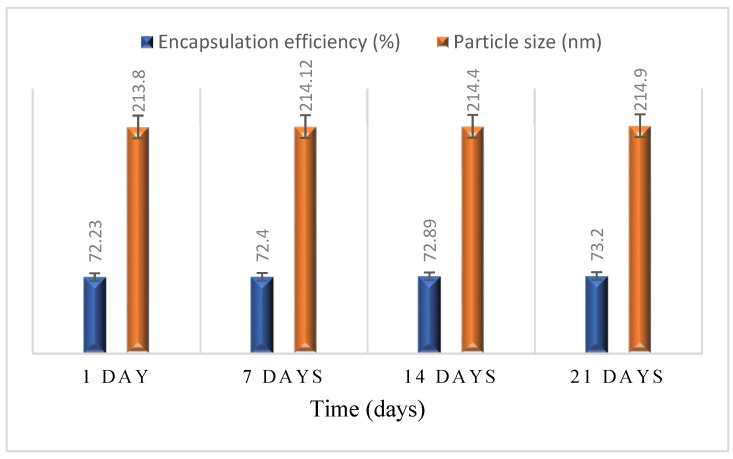
Stability studies of MTX loaded AEM/CS NPs over a time duration of 21 days in mean ± S.D (*n* = 3).

**Table 1 ijms-23-02725-t001:** % Encapsulation efficiency (%EE) of MTX loaded AEM-CS NPs in mean ± standard deviation (S.D.)/*n* = 3.

Formulation	AEM (*w/v* %)	CS (*w/v* %)	%Encapsulation Efficiency
F1	0.02	0.01	42.1 ± 1.2
F2	0.01	0.01	72.2 ± 2.0
F3	0.01	0.02	53.4 ± 2.1

**Table 2 ijms-23-02725-t002:** Drug content and percent yield of MTX loaded AEM-CS NPs in mean ± S.D. (*n* = 3).

Formulation	% Drug Content	% Yield
F1	84 ± 1.3	75.7 ± 1.3
F2	94.5 ± 1.6	50 ± 1.2
F3	81.2 ± 1.2	84.3 ± 0.8

**Table 3 ijms-23-02725-t003:** Characterization of MTX loaded AEM-CS NPs.

Sample	AEM: CS (Mass Ratio)	Z-Average (nm)	Polydispersity Index	ζ-Potential (mV)
F1	2:1	238.4	0.485	−9.1
F2	1:1	213.8	0.279	+11.4
F3	1:2	254.2	0.361	+22.7

**Table 4 ijms-23-02725-t004:** (**a**): Modelling and release kinetics of nanoparticles at pH 5.5. (**b**): Modelling and release kinetics of nanoparticles at pH 7.4.

(**a**)											
Nanoparticles	Zero Order		First Order		Higuchi Model		Korsmeyer-Peppas Model		Peppas-Sahlin		
	R^2^	K_0_	R^2^	K_1_	R^2^	K_H_	N	K_r_	R^2^	K_d_	K_r_
F1	0.3098	1.989	0.582	0.031	0.958	9.62	0.45	12.02	0.985	10.34	2.64
F2	0.394	2.333	0.863	0.019	0.974	11.22	0.475	13.44	0.993	10.736	3.16
F3	0.7851	1.472	0.863	0.019	0.9573	6.852	0.58	5.444	0.953	5.992	1.074
(**b**)											
Nanoparticles	Zero Order		First Order		Higuchi Model		Korsmeyer-Peppas Model		Peppas-Sahlin		
	R^2^	K_0_	R^2^	K_1_	R^2^	K_H_	N	K_r_	R^2^	K_d_	K_r_
F1	0.430	2.418	0.670	0.034	0.916	9.93	0.37	9.92	0.899	10.505	4.7
F2	0.606	1.827	0.746	0.023	0.954	6.97	0.32	6.56	0.909	5.37	6.38
F3	0.403	2.891	0.712	0.044	0.948	11.25	0.36	12.23	0.923	11.02	6.98

## Data Availability

Not applicable.

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
