# Peer review of "pH Responsive Abelmoschus esculentus Mucilage and Administration of Methotrexate: In-Vitro Antitumor and In-Vivo Toxicity Evaluation"

_ijms, 2022, doi:10.3390/ijms23052725_

Round 1

Reviewer 1 Report

The subject of the manuscript entitled ‘pH responsive Abelmoschus esculentus mucilage and Chitosan ionically cross-linked Nanoparticles for Oral Administration of Methotrexate: In-vitro antitumor and in-vivo toxicity evaluation’ is of interest while not being significantly novel and reflects the journal’s scope. In general, the journal’s guidelines regarding article structure have been followed. However, certain claims/conclusions are not fully supported by the data and certain mentioned results seem to not be presented in figures. Some of the figures should be redesigned so as data and labels are clearly provided. Moreover, figure legends should be more detailed and concise. Most importantly, methodology should be described in more detail. Additional experiments are proposed in order to confirm paper's main premise i.e. the the cytotoxicity of the NPs against cancer cells. Finally, the Discussion section can be more focused.

More specific comments follow:

It is difficult to follow the narrative of the manuscript mainly because of the organization of the different sections. Mat and Methods should precede Results or a brief description should be added so that the reader can understand what Formulations 1, 2 or 3 are before diving in their comparative analysis. Results presented in Table 3 and Figure 3 should precede Tables 1 and 2 and Figures 1 and 2 for the same reason.

The authors should provide more details in figure legends. For example in Tables 1 and 2, it should be mentioned what each Formulation is and what is being encapsulated, in Table 2  the units (% of what) shouls be added,  CDR should be defined in Figure 4, a description of the assay should be added in Figure 6 as well as a description of what ‘blank NPs’ stands for.

Figure 6A is of low quality and does not provide additional useful information. Moreover, the graph in Figure 6B lacks error bars/statistics.

The in vivo study described in 3.5.7. In-vivo acute toxicity of AEM-CS based blank nanoparticles, is not described adequately, in fact the study design is missing. What was administered to the animals, for how many doses of what concentration and what was the route of administration? Moreover, the authors state in line 355: The acute toxicity of both polymers i.e. AEM and CS was tested by maximal tolerance dose (MTD) technique. As presented in results there is only one Test group -could the authors explain what did they test? Also, for the determination of the maximum tolerance dose, multiple treatment groups should be employed for escalating doses. The authors should provide additional information and/or restate certain claims.

Tables and graphs in Figure 5 should be corrected and redesigned so as to accurately and clearly represent the results. There are several mistakes that make it hard to understand the results (labels, numbers that need to be superscripted etc). Also, more information regarding the study design, numbers of animals, statistical analysis etc should be provided in Figure legend.

It is stated throughout the manuscript that the F2 formulation was tested regarding its cytotoxicity in MCF7 breast cancer cells and in non-cancerous cells (lines 84 and line 247). However, the results presented in Figure 6 concern only one cell line (MCF7).  More info should be provided regarding the procedure and the steps followed for the in vitro cytotoxicity assay. Also, additional results should be presented or the authors must change these claims.

In paragraph 3.5.4 the method for determining % yield and drug content is barely described. The authors provide a reference but, again, a more detailed description of the steps followed and the instruments used should be added.

How were the hematological and biochemical profiles of the animals determined? Again, instruments, kits and methodology are important and shoulf be mentioned.

Regarding statistical analysis, besides one-way ANOVA, what was the post hoc test the authors used?

The cytotoxic effect of NPs against cancer cells is moderate. The highest cell growth inhibition observed according to Figure 6B is around 30% but the authors used a NPs concentration and a short time period for the incubation. I would suggest that the authors conduct more cytotoxicity studies, for higher concentrations and extended  incubation periods, preferably in more than one cancer cell lines. Moreover, since the investigation of the enhanced bioavailability of MTX due to the proposed nanoformulation/carrier is among the aims of this study, it would be interesting to have a comparative cytoxicity analysis assaying both the proposed NPs' system and the equivalent pure MTX concentration.

There are various pH-responsive nanoformulations that have been proposed for the effective delivery of MTX in tumor cells (e.g. Rahman, M., Khan, J.A., Kanwal, U. et al. Methotrexate-loaded PEGylated gold nanoparticles as hemocompatible and pH-responsive anticancer drug nanoconjugate. J Nanopart Res 23, 195 (2021), Najafipour, Aylar, et al. "MTX-loaded dual thermoresponsive and pH-responsive magnetic hydrogel nanocomposite particles for combined controlled drug delivery and hyperthermia therapy of cancer." Molecular Pharmaceutics 18.1 (2020): 275-284, Zhang, Keliang, et al. "Construction of a pH-responsive drug delivery platform based on the hybrid of mesoporous silica and chitosan." Journal of Saudi Chemical Society 25.1 (2021): 101174, Carrillo-Castillo, Teresa Darlen, et al. "pH-responsive polymer micelles for methotrexate delivery at tumor microenvironments." e-Polymers 20.1 (2020): 624-635). Could the authors discuss a comparison between their system and the previously proposed types of nanoparticles and highlight the novelty of their technology and its advantages/disadvantages if any?

And some minor issues:

There are few wording errors or word choices that make the text prone to misunderstanding.  For example, line 17: The rapid procession in biomaterial nanotechnology comprehends the potential of nontoxic and potent polysaccharide…, or line 47: In this smart nanometric carriers contour, the tide has shifted largely… , line 72: has been in clinical instrumental use…, line 75: However, its therapeutic efficacy is often seriously compromised by its undiscerning systemic distribution leading…, line 368: The cell line was developed in DMEM, among other instances. 

Acronyms or abbreviations must be defined upon first use, please add an explanation where appropriate e.g. in abstract, line 23: PDI

Line 225: LD50 is NOT the therapeutic index.

The concentrations of the NPs are in ng/μl (line 372) or in ng/ml (Figure 6B)?

Lines 373-374: The following phrase should be rewritten; the reader cannot comprehend the steps followed: MTT with a concentration of 5 mg/ml and 100 μl was added to each well, followed by additional 4 h incubation, with subsequent removal of the culture media with DMSO in each well.

Line 375: 570 nm do not fall in the UV range.

Line 57: Abelmoschus esculentus mucilage (AEM), natively known as okra, is an anionic polysaccharide that includes a dominant… the plant and not the mucilage is known as okra, please correct.

Lines 66-67: The tailoring of CS-NPs with anionic polysaccharide i.e. AEM has shown to decrease macrophage uptake enhancing the efficacy of cargo enclosed within. Please add reference.

Line 86: We also exposed MCF-7 cell lines and non-cancerous cells to MTX/AEM-CS NPs for cell viability evaluation which showed that MTX/AEM-CS NPs could be a potential alternative to commercial anticancer drugs, increasing patient compliance in clinical practice. It is not clear how the outcomes of this study could affect patient compliance, maybe the authors need to further discuss this claim.

In Results, lines 97-100: The results showed that AEM-CS based formulations would enable the chemotherapeutics to be used to their full therapeutic potential in controlled ways with alleviating the toxicities associated with them. Moreover, the encapsulation of MTX with AEM mucilage would enable these nanoparticles to be delivered orally as it would provide protection from gastric acidic environment. In the second sentence I suppose the authors refer to the AEM-CS nanoparticles however the authors have shown that these nanoparticles release the MTX cargo in acidic pH, therefore, how would it be possible to offer protection from gastric acids? 

In certain instances text is not super/subscripted where it should have been, e.g. lines 108 and 115 cm-1, line 293 40C, graphs in Figure 5, etc.

Line 123: Figure 1 (a) should change to Figure 2 (a).

In Figure 2 and in the respective text, can the authors define what they specifically mean by 'physical mixture'?

Line 292: drug concentration is mentioned as 0.0.1 % w/v, please correct.

Leave a gap between the value and the units.

In Figure 7, what does the axis stand for, what are the units? In general all the units and adequate informations must be provided for all the graphs.

In equations 2 and 3 some words are mispelled/there are letters missing.

The mouse strain employed should me mentioned.

Author Response

We appreciate the suggestion and comments from the reviewers and have tried our best to incorporate the changes in the manuscript with tracking changes turned on. The language and format of manuscript have been modified.  The mistakes and errors you suggested have been corrected. Please check the attached document

Reviewer 2 Report

This manuscript reports a potentially useful drug delivery system (DDS) for cancer therapy, but serious flaws on the manuscript must be corrected before publication.

There are major issues that need to be addressed, specifically:

  1. Table 1 needs correction: EE% values are presented with two decimal places, but the uncertainty (SD values) are in the unit digit. So, the decimal places have no physical meaning.
  2. DLS results: a PDI range of 0.279–0.485 clearly indicates polydisperse nanoparticles. PDI above 0.3 means highly polydisperse nanoparticles. Please, correct this in Results and Discussion and also in Abstract.
  3. Figure 3: SEM images of particles AEM:CS (2:1) and AEM:CS (1:2) are missing and must be added. Moreover, size histograms obtained from SEM images must be included and the results compared to the results obtained from DLS measurements.
  4. Figure 4: How many replicas were performed? No error bars are presented. At least, a triplicate assay must be presented. What are the units of time in x-axis?
  5. The equations of the several release models must be presented, with the meaning of the variables (e.g. kd, kr,…)
  6. Table 4 is unformatted, the names of the models are not aligned with the corresponding values, subscripts and superscripts were ignored. Units of k are missings.
  7. Cytotoxicity assays were only performed for F2 nanoparticles. Therefore, similar studies for F1 and F3 nanoparticles are missing and must be added. Moreover, other tumor cell lines and not only MCF7 (a very susceptible cell line) must be used. In fact, the decrease in cell viability is very poor, pointing only to a limited therapeutic effect of this DDS.
  8. Figure 7. Again, error bars are missing. How many measurements were performed? Please add the error bars.
  9. The conclusions are highly speculative, because only one cancer cell line (MCF7, known to be very susceptible) was used. The study must be completed with, at least, one more cancer cell line and several formulations must be tested. Also, the DDS must be assayed also in the corresponding normal (non-tumor) cells.

There are several mistakes along the manuscript (examples below), and unclear sentences, clearly showing that this is not a final version of the manuscript. Some examples below:

equation 1 “free dru”

Equation 2: “weigh” “weig”

Equation 3: “Weig”

“Lastly, the diffusion of the drug is constant mainly occurs via pores” (line 205)

Krosmeyer (line 207)

“Whereas the kinetic release results at pH 7.4 showed that diffusion was mainly followed for drug release.” (line 213)

25C± 2C (line 379) - missing degrees 

Author Response

We appreciate the suggestion and comments from the reviewers and have tried our best to incorporate the changes in the manuscript with tracking changes turned on. The language and format of manuscript have been modified.  The mistakes and errors you suggested have been corrected. Find below the attached document please.

Reviewer 3 Report

The topic of this article is interesting, the authors presenting the obtaining and characterization of methotrexate polysaccharide nanoparticles based on Abelmoschus esculentus mucilage and chitosan. They assessed the methotrexate release profile, the in vivo biocompatibility in mice and evaluated the nanoparticles effects on the breast cancer MCF-7 cell and on non-cancerous culture cells viability.

After reading the manuscript, the following doubts and suggestions have arisen:

The results obtained should be compared with those achieved by other researchers and discussions should be significantly detailed. This section should be more complete, providing supplementary background about the recent communicated data about other nano-systems containing methotrexate.

Literature analysis reveals various communicated data in the field should be cited, for example:

Wang J et al. Lactobionic acid-modified thymine-chitosan nanoparticles as potential carriers for methotrexate delivery. Carbohydr Res. 2021 Mar; 501:108275. 

Shakeran Z et al. Biodegradable nanocarriers based on chitosan-modified mesoporous silica nanoparticles for delivery of methotrexate for application in breast cancer treatment. Mater Sci Eng C Mater Biol Appl. 2021 Jan;118:111526. 

Nur Gunel Selvi et al. Synthesis of Methotrexate Loaded Chitosan Nanoparticles and in vitro Evaluation of the Potential in Treatment of Prostate Cancer, Anti-Cancer Agents in Medicinal Chemistry 2016; 16(8): 1038-1042.

Jhaveri J et al. Chitosan Nanoparticles-Insight into Properties, Functionalization and Applications in Drug Delivery and Theranostics. Molecules 2021, 26, 272.

Dehshahri A et al. New Horizons in Hydrogels for Methotrexate Delivery. Gels 2021, 7, 2.

Sankha Bhattacharya. Methotrexate-loaded polymeric lipid hybrid nanoparticles (PLHNPs): a reliable drug delivery system for the treatment of glioblastoma, Journal of Experimental Nanoscience, 2021; 16:1, 344-367, 

Carrillo-Castillo T-D et al. Thermosensitive hydrogel for in situ-controlled methotrexate delivery. e-Polymers, vol. 21, no. 1, 2021, pp. 910-920.

Surve C et al. Formulation and qbd based optimization of methotrexate-loaded solid lipid nanoparticles for an effective anti-cancer treatment. International Journal of Applied Pharmaceutics, 2021, 13(5), 132–143.

Mehreen Rahman et al. Methotrexate-loaded PEGylated gold nanoparticles as hemocompatible and pH-responsive anticancer drug nanoconjugate. Journal of Nanoparticle Research, 2021, 8.

Some other aspects were found in this manuscript:

- different fonts were used in the text and in same figures;

  • missing information (city, country) about the companies Sigma (line 271) and Fisher Chemicals (line 274);
  • missing information about mice species;
  • missing information (city, country) about the companies producing some devices used in the research (i.e. line 154 - DLS; line 311 - SEM apparatus; line 338 - USP dissolution apparatus);

- the abbreviations should be mentioned in the text (i.e. MTT, DMSO, ALT, AUS, ALP, AST, Hb, WBCs, RBCs, S.D., S.E.M.)

- the authors mentioned they used S.D. in statistical analysis, but presented S.E.M. in the results.

- spelling check of the text is mandatory.

- the authors should upgrade the references;

- English including grammar, style and syntax, should be extensively improved through the professional help from English Editing Company for Scientific Writings.

Author Response

We appreciate the suggestion and comments from the reviewers and have tried our best to incorporate the changes in the manuscript with tracking changes turned on. The language and format of manuscript have been modified.  The mistakes and errors you suggested have been corrected. Please check the attached document for responses. 

Round 2

Reviewer 1 Report

The authors have tried to address most of the issues raised and conducted additional experiments. However, the manuscript remains difficult to follow. There are many wording/grammar/syntax errors. The narrative keeps being complex with backs and forths. Figures need to be improved; there are mistakes in legends, axes titles etc, certain labels are too small to be read and certain described results are not presented (eg the 24-h cell viability study). Moreover, there are conclusions that are not supported by the results; eg line 347: MTX displayed similar cell proliferation inhibition potential but to slightly lesser extent -based on the results presented, this is not the case. I' d suggest a thorough revisiοn of the manscript for languange and clarity with the help of a native English speaker.

Author Response

pH responsive Abelmoschus esculentus mucilage and Chitosan ionically cross-linked Nanoparticles for Oral Administration of Methotrexate: In-vitro antitumor and in-vivo toxicity evaluation

We appreciate the suggestion and comments from the reviewers and have tried our best to incorporate the changes in the manuscript with tracking changes turned on. The language and format of manuscript have been thoroughly revised. The mistakes and errors you suggested have been corrected.

The point to point response to reviewers is as follows 

Sr.

No.

Reviewer I

Response to comments

1.

There are many wording/grammar/syntax errors. The narrative keeps being complex with backs and forths. Figures need to be improved; there are mistakes in legends, axes titles etc, certain labels are too small to be read and certain described results are not presented (eg the 24-h cell viability study).

Thank you for the comment. We have redrawn figures with corrections in their legends and axes tiles. Moreover the 24h cell viability results have been added too as Figure 6A.

2.

There are conclusions that are not supported by the results; eg line 347: MTX displayed similar cell proliferation inhibition potential but to slightly lesser extent -based on the results presented, this is not the case.

Thank you for the valuable suggestion. We have revised the manuscript especially the points in the cytotoxicity studies you have mentioned (Heading 2.8).   

Reviewer 2 Report

The authors addressed most of the comments with additional experiments. However, the manuscript has still parts very hard to understand, e.g.:

Line 88: However, its therapeutic efficacy is often seriously compromised by its insignificant systemic distribution leading to side effects such as alopecia, nausea, body aches, hepatotoxicity and myelosuppression [21].

Lines 95-97: MXT means what?

Micro is represented by “u”, e.g., ug/ml

“The % EE was observed to be decreasing with an upsurge in either of polymers concentration.”

I suggest revision of English with English Editing Services.

Author Response

pH responsive Abelmoschus esculentus mucilage and Chitosan ionically cross-linked Nanoparticles for Oral Administration of Methotrexate: In-vitro antitumor and in-vivo toxicity evaluation

We appreciate the suggestion and comments from the reviewers and have tried our best to incorporate the changes in the manuscript with tracking changes turned on. The language and format of manuscript have been thoroughly revised. The mistakes and errors you suggested have been corrected.

The point to point response to reviewers is as follows 

Sr.

No.

Reviewer II

Response to comments

1.

Line 88: However, its therapeutic efficacy is often seriously compromised by its insignificant systemic distribution leading to side effects such as alopecia, nausea, body aches, hepatotoxicity and myelosuppression [21].

Thank you for the comment. The sentence has been rewritten to make its meaning more clear (Line 90).

2.

MXT means what?

Micro is represented by “u”, e.g., ug/m

Thank you. The mistakes have been corrected through out the manuscript.

“The % EE was observed to be decreasing with an upsurge in either of polymers concentration.”

Thank you for the suggestion. We have reconstituted the mentioned sentence (Line 266)

Reviewer 3 Report

I received the manuscript in which the authors made the changes according to the reviewers recommendations and suggestions. It consider it could be accepted for publication in this journal, but I propose to have the manuscript checked by a native English speaking person.

Author Response

We cherish the suggestion and comments from the reviewer and have tried our best to incorporate the changes in the manuscript with tracking changes turned on. The language and format of manuscript has been thoroughly revised. The mistakes and errors you suggested have been corrected.